# Cardiovascular Magnetic Resonance Demonstrates Myocardial Inflammation of Differing Etiologies and Acuities in Patients with Genetic and Inflammatory Myopathies

**DOI:** 10.3390/jcm12041575

**Published:** 2023-02-16

**Authors:** George Markousis-Mavrogenis, Antonios Belegrinos, Aikaterini Giannakopoulou, Antigoni Papavasiliou, Vasiliki Koulouri, Nikolaos Marketos, Eleftheria Patsilinakou, Fotini Lazarioti, Flora Bacopoulou, Clio P. Mavragani, George P. Chrousos, Sophie I. Mavrogeni

**Affiliations:** 1Olympic Diagnostic/Research Center, 17674 Athens, Greece; 2Onassis Cardiac Surgery Center, 17674 Athens, Greece; 3Faculty of Medicine, National and Kapodistrian University of Athens, 15772 Athens, Greece; 4Cardiology Clinic, Aghia Sophia Children’s Hospital, 11527 Athens, Greece; 5Iaso Children’s Hospital, 15123 Athens, Greece; 6Department of Physiology “Molecular Physiology and Clinical Applications Unit”, Faculty of Medicine, National and Kapodistrian University of Athens, 11527 Athens, Greece; 7University Research Institute for Maternal and Child Health and Precision Medicine, School of Medicine, National and Kapodistrian University of Athens, 11527 Athens, Greece; 8Center for Adolescent Medicine and UNESCO Chair in Adolescent Health Care, First Department of Pediatrics, School of Medicine, National and Kapodistrian University of Athens, Aghia Sophia Children’s Hospital, 11527 Athens, Greece; 9Attikon Hospital, 12462 Athens, Greece

**Keywords:** T1 mapping, T2 mapping, ECV, LGE, subclinical

## Abstract

Introduction. Myopathies are heterogeneous neuromuscular diseases of genetic and/or inflammatory etiology that affect both cardiac and skeletal muscle. We investigated the prevalence of cardiac inflammation in patients with myopathies, cardiovascular symptoms, and normal echocardiography using cardiovascular magnetic resonance (CMR). Methods. We prospectively evaluated 51 patients with various genetic (n = 23) and inflammatory (n = 28) myopathies (median age, IQR: 12 (11–15) years, 22% girls; 61 (55–65) years, 46% women, respectively) and compared their CMR findings to corresponding age- and sex-matched controls (n = 21 and 20, respectively) and to each other. Results. Patients with genetic myopathy had similar biventricular morphology and function to healthy controls but showed higher late gadolinium enhancement (LGE), native T1 mapping, extracellular volume fraction (ECV), and T2 mapping values. Collectively, 22 (95.7%) patients with genetic myopathy had a positive T1-criterion and 3 (13.0%) had a positive T2-criterion according to the updated Lake Louise criteria. Compared with healthy controls, patients with inflammatory myopathy showed preserved left ventricular (LV) function and reduced LV mass, while all CMR-derived tissue characterization indices were significantly higher (*p* < 0.001 for all). All patients had a positive T1-criterion, and 27 (96.4%) had a positive T2-criterion. A positive T2-criterion or T2-mapping > 50 ms could discriminate between patients with genetic and inflammatory myopathies with a sensitivity of 96.4% and a specificity of 91.3% (AUC = 0.9557). Conclusions. The vast majority of symptomatic patients with inflammatory myopathies and normal echocardiography show evidence of acute myocardial inflammation. In contrast, acute inflammation is rare in patients with genetic myopathies, who show evidence of chronic low-grade inflammation.

## 1. Introduction

The myopathies are a heterogeneous group of neuromuscular diseases, with the common disease characteristic being muscular weakness due to disease processes at the level of the muscle fiber [1]. Myopathies can have diverse etiologies and can be subdivided into two broad categories, namely, myopathies due to genetic defects [2] and myopathies of inflammatory and/or autoimmune etiology [3,4].

Although skeletal muscles are most often affected, myopathies may also involve cardiac muscle, and this may occur prior to, simultaneously with, or after the onset of skeletal muscle disease [5,6]. Notably, because of the often clinically silent onset and progression of the latter, myopathies can also be life-threatening conditions [5,6]. The recognition of this important fact is reflected by their characterization as rare dilated cardiomyopathies according to the Classification of Rare Cardiovascular Diseases (RCD Classification) [7]. As such, patients with myopathies, irrespective of cause, should also be evaluated for concurrent cardiac disease [8].

Currently, the primary diagnostic tools capable of evaluating the heart in this context are echocardiography and cardiovascular magnetic resonance (CMR). Echocardiography is widely available with widespread expertise among cardiologists, can be performed at the bedside, and is cost-effective. However, it is operator-dependent, and its efficacy can be limited by poor acoustic windows. CMR, on the other hand, is the diagnostic gold standard for the evaluation of ventricular/atrial volumes, wall motion and systolic function of both ventricles, importantly without the need for ionizing radiation. Due to its unique ability to perform tissue characterization, it offers incremental diagnostic/prognostic information in heart failure with preserved (HFpEF) or reduced ejection fraction (HFrEF) [9,10,11]. Specifically, the evaluation of late gadolinium-enhanced (LGE) images is considered the gold standard for the assessment of replacement fibrosis [12].

Although CMR has been used previously to investigate myocardial inflammation both in patients with inflammatory myopathies [13] and genetic myopathies [6,14], to our knowledge, no previous study has performed a comparison of CMR findings between these two sub-groups of myopathies. In addition, little is known regarding the presence of myocardial inflammation and fibrosis in symptomatic patients with myopathies and normal routine clinical and echocardiographic evaluation. Thus, we hypothesized that CMR could provide important insights about myocardial involvement in patients with myopathies with atypical cardiac symptoms and normal routine clinical and echocardiographic evaluation. This study had the following aims:(1)To examine if CMR can identify inflammatory or fibrotic loci in patients with myopathies and atypical cardiac symptoms with otherwise normal routine clinical and echocardiographic evaluation.(2)To assess if CMR findings could provide insights regarding the origin of myocardial abnormalities in patients with myopathies.(3)To compare the tissue characteristics of the myocardium in patients with genetic and inflammatory myopathies.

## 2. Methods

### 2.1. Study Participants

A total of 51 patients diagnosed with a myopathy (23 with various genetic myopathies and 28 with various inflammatory myopathies) that were referred for cardiology evaluation due to atypical cardiac symptoms (chest discomfort, mild shortness of breath or palpitations) and normal routine clinical and echocardiographic evaluation were prospectively evaluated using CMR. The CMR findings of each group of patients were compared with a corresponding age- and sex-matched healthy control group (21 and 20 healthy controls for the genetic and inflammatory myopathy group, respectively). The study protocol was approved by the Olympic Diagnostic/Research Center ethics committee (protocol number 2, 4 November 2022), and all participants provided written informed consent before inclusion in the study.

### 2.2. CMR Protocol

All individuals included in the study underwent a CMR evaluation, including biventricular function evaluation and T2 and T1 imaging assessment for the evaluation of myocardial inflammation and fibrosis using a 3.0T system (Skyra, Siemens Healthineers, Erlangen, Germany).

### 2.3. Biventricular Function Analysis

The CMR examination included standard functional imaging using a high spatial and temporal resolution and breath-held balanced steady-state precession (bSSFP) cine sequence [15,16] with the following acquisition parameters: 58° flip angle, rate-3 parallel imaging, matrix size 256 × 192, pixel size 1.6 mm × 1.6 mm, slice thickness 6 mm, BW 977 Hz/Px, TE/TR 1.4 ms/3.3 ms echo spacing, and a temporal resolution of 32.5 ms. All cine imaging included the entire LV from base to apex using short-axis slices.

### 2.4. T2 Imaging for Oedema

Black-blood short tau T2W images (STIR-T2) for edema detection were also acquired. For T2 mapping, data were acquired in basal, mid-ventricular, and apical short-axis planes using a T2-prepared single-shot SSFP technique, as was previously described [17].

### 2.5. T1 Imaging for Inflammation/Fibrosis

T1-weighted spin-echo early gadolinium-enhanced (EGE) and phase-sensitive inversion recovery late gadolinium-enhanced (LGE) images were acquired after intravenous injection of gadobenate dimeglumine contrast medium (Gadoteric acid, Cyclolux, VIANEX), as described previously [18].

Myocardial T1 mapping measurements were acquired using a modified Look-Locker inversion recovery (MOLLI) sequence with motion correction (MOCO) [19] in basal, mid-level, and apical ventricular short-axis slices and were performed with electrocardiographic gating and breath holding. Pre- and post-contrast T1 mapping was acquired with a 5(3 s)3 and a 4(1 s)3(1 s)2 MOLLI scheme, respectively. Typical native T1 imaging parameters were: non-selective inversion pulse, bSSFP single shot readout with a 20° excitation flip angle, 7/8 partial Fourier and rate-2 parallel imaging, matrix size 192 × 132, pixel size 1.9 mm × 1.9 mm, slice thickness 8 mm, BW 1085 Hz/Px, minimum inversion time (TI) of 100 ms incremented by 80 ms, and TE/TR 1.01 ms/2.44 ms echo spacing. Typical post-contrast T1 imaging parameters were: non-selective inversion pulse, bSSFP single-shot read out with a 20° excitation flip angle, 7/8 partial Fourier and rate-2 parallel imaging, matrix size 192 × 164, pixel size 1.9 mm × 1.9 mm, slice thickness 8 mm, BW 1085 Hz/Px, minimum TI of 100 ms with 80 ms increments, and TE/TR 1.01 ms/2.44 ms echo spacing. Approximately 18 min (17 ± 5 min) after contrast injection, post-contrast T1 mapping was performed at slice locations matched to the pre-contrast acquisition. The MOCO T1 (pre- and post-contrast) maps were generated by the scanner and later used for calculating ECV maps. All patients provided a recent measurement of hematocrit to be used in calculating the subject-specific ECV [20].

### 2.6. Post-Processing Analyses

Two experienced clinicians (SM, FL) with >20 years of experience evaluated the images using the Syngo Siemens protocol. The following functional parameters were calculated: LV-RV end-systolic volume (LVESV, RVEDV) and end-diastolic volume (LVEDV, RVEDV), ejection fraction (LVEF, RVEF), and LV mass (LVM). A normal LVEF was defined as LVEF ≥ 55% [19]. Additionally, the presence or absence of LGE was identified according to the American Heart Association (AHA) 17-segment model [21]. Pre- and post-contrast T1 maps were combined with each patient’s hematocrit to calculate an ECV map [22]. A region of interest (ROI) encompassing the basal, mid-level, and apical LV myocardium was manually selected and analyzed. Values from each ROI were selected, and a mean value was calculated to represent the native and post-contrast T1 mapping values of the myocardium.

T1-mapping was performed using a 5-3-3 modified Look-Locker inversion recovery (MOLLI) technique with a scheme on three representative short-axis positions (base, middle, apex) before and 15 min after contrast-media administration. T2-mapping was performed on the same three left ventricular (LV) short axes using a black-blood-prepared, navigator-gated, free-breathing hybrid gradient (echo planar imaging) and spin-echo multi-echo sequence.

### 2.7. CMR Data Analysis

Global myocardial inflammation was assessed in STIR-T2 images by calculating the T2 signal intensity ratio as the signal intensity of myocardium divided by the signal intensity of the skeletal muscle (T2 ratio) [18]. Global relative enhancement was calculated by measuring myocardial signal intensity on pre- and post-contrast T1-weighted spin-echo images relative to skeletal muscle [23]. The presence and pattern of LGE lesions were quantitatively expressed as % of LVM by consensus agreement of two experienced observers. Intra- and inter-observer agreement was 0.88 and 0.85, respectively.

Color-coded T1 and T2 maps were generated based on inline-generated, motion-corrected raw images using built-in software in three matching short-axis slices. Motion-corrected T1 maps were examined for quality in raw T1 images, T1 maps and T2 maps. Endocardial and epicardial contours were manually drawn by two experienced observers. Global T1, ECV, and T2 values were calculated. Before the CMR examination, the hematocrit was determined in all subjects, allowing the calculation of ECV in conjunction with native and post-contrast T1-mapping measurements using a previously described equation [18]. T2 results were obtained by fitting a two-parameter, intensity-weighted exponential model [18]. The mapping results, together with other CMR indices, were evaluated according to a previously described methodology [24].

### 2.8. Evaluation of Myocardial Inflammation

To evaluate the presence of myocardial inflammation, we made use of the updated Lake Louise criteria (LLc), which are comprised of a T1-based criterion and a T2-based criterion [24]. A positive T1-based criterion was defined as the presence of any pathologic T1-based index (EGE > 4, LGE > 0% of LVM, native T1 mapping > 1250 ms, ECV > 28%), and a positive T2-based criterion was defined as the presence of any pathologic T2-based index (T2 ratio> 2, T2 mapping > 50 ms). Positivity for both criteria was considered to suggest a high probability of acute myocardial inflammation, while positivity for either criterion could suggest the presence of acute myocardial inflammation but with lower specificity [24].

### 2.9. Statistical Analysis

Data were analyzed using R-Studio (R v.4.1.2). Body surface area (BSA) was calculated using the Du Bois method, and ventricular volumes and LVM were indexed by dividing the values by the calculated BSA. The normality of continuous variables was examined visually using histograms and Q-Q plots, and all were found to be not normally distributed. Thus, comparisons between groups were performed with Mann–Whitney tests for continuous variables and with Chi-square tests for binary/categorical variables. Continuous variables are presented as median (interquartile range), and binary/categorical variables are presented as numbers (percentage). Statistical significance was considered for *p* ≤ 0.05. Boxplots with individual data points were used for visualizing different CMR indices stratified by group. Where mentioned, sensitivity and specificity were calculated based on the true/false positive and true/false negative values from the corresponding contingency table. Receiver operating characteristic curves were plotted based on univariable logistic regression models.

## 3. Results

### 3.1. Comparison of Patients with Genetic Myopathies to Matched Healthy Controls

The majority of patients with genetic myopathies were diagnosed either with Duchenne muscular dystrophy (8 (34.8%)) or Becker muscular dystrophy (7 (30.4%)). Patients with genetic myopathies did not have significant differences in biventricular morphology, function, or LVM (Table 1). Compared with their matched controls, patients with genetic myopathies had, on average, significantly higher LGE (0% (0, 0) vs. 6.0% (3.0, 8.0), *p* < 0.001), native T1 mapping (1170.0 (1158.00, 1182.0) vs. 1250.0 (1190.0, 1293.5), *p* < 0.001), and ECV values (23.0 (23.0, 25.0) vs. 28.0 (26.5, 30.0), *p* < 0.001). Similarly, compared to controls, patients with genetic myopathy had increased T2 mapping (40.0 (37.0, 43.0) vs. 46.00 (43.00, 48.00), *p* < 0.001) but not T2 ratio (*p* = 0.646). Nevertheless, when investigating the number of participants with abnormal CMR values, only LGE, native T1 mapping and ECV showed significantly higher percentages in the genetic myopathy group (19 (82.6%), 10 (43.5%), and 11 (47.8), respectively, *p* ≤ 0.002 for all; matched controls had no occurrences of abnormal CMR values). Only 2 (8.7%) patients showed abnormal T2 ratio/T2 mapping values (*p* = 0.51). No controls met any of the updated LLc, while 22 (95.7%) and 3 (13.0%) of patients with genetic myopathies met either the T1-based or T2-based criterion, respectively (*p* < 0.001 and *p* = 0.265, respectively). By extension, 3 (13.0%) patients also met both criteria, suggesting a high probability of myocardial inflammation.

### 3.2. Comparison of Patients with Inflammatory Myopathies to Matched Healthy Controls

The majority of patients with inflammatory myopathies were diagnosed either with polymyositis (10 (35.7%)) or dermatomyositis (10 (35.7%)). Similar to the previous comparison, patients with inflammatory myopathies did not show significant differences in age or sex compared with their matched control group (Table 2). Patients with inflammatory myopathies had a significantly higher LVEF and RVEDVi compared with their matched controls (68.2% (61.2, 71.9) vs. 58.4% (54.1, 60.2), *p* < 0.001 and 68.0 (56.1, 77.3) vs. 53.2 (50.9, 61.5), *p* = 0.007, respectively). In contrast, patients with inflammatory myopathy had a significantly lower indexed LVM compared with their matched controls (52.0 (48.6, 55.2) vs. 57.5 (54.8, 65.2), *p* = 0.001). Compared with their matched controls, patients with inflammatory myopathy showed significantly elevated values in all T1-based indices (EGE, LGE, native T1 mapping, ECV) and T2-based indices (T2 ratio, T2 mapping) (*p* < 0.001 for all). None of the controls met any of the updated LLc, while all 28 patients with inflammatory myopathy met the T1-based criterion, and 27 (96.4%) met the T2-based criterion. Collectively, 27 (96.4) patients met both criteria, suggesting a high probability of myocardial inflammation.

### 3.3. Comparison of Patients with Genetic and Inflammatory Myopathies

The comparison of patients with genetic and inflammatory myopathies is presented in Table 3. Apart from the expected significant difference in age, patients with inflammatory myopathies again had significantly higher LVEF compared to those with genetic myopathies (*p* < 0.001). Regarding T1-based indices, although there were no significant differences in LGE and ECV between the groups, patients with inflammatory myopathies had significantly higher EGE and native T1 values (*p* < 0.001 and *p* = 0.004, respectively). Both groups also showed considerable differences in T2-based indices, with patients with inflammatory myopathies having much higher values of both T2 ratio and T2 mapping (2.80 (2.50, 3.20) vs. 1.50 (1.20, 1.80), *p* < 0.001 and 58.0 ms (55.0, 61.3) vs. 46.0 ms (43.0, 48.0), *p* < 0.001, respectively). When cut-off points for normal values were examined, the proportion of patients with inflammatory myopathies that had abnormal T1- or T2-based indices was significantly higher compared with patients with genetic myopathies, with the exception of LGE (*p* = 0.379). Although the large majority of patients in either group met the T1-based LLc (*p* = 0.921), significantly more patients with inflammatory myopathies met the T2-based LLc (27 (96.4%) vs. 3 (13.0%), *p* < 0.001).

Comparisons of CMR indices between all of the examined groups are presented visually in Figure 1 and Figure 2, separated according to T1-based and T2-based indices. Both through visual inspection as well as statistical testing, it became apparent that the T2-based criterion and T2-mapping could optimally discriminate between patients with genetic and inflammatory myopathies. An ROC curve was plotted for this binary outcome and revealed an area under the curve of 0.9557 (Figure 3), suggesting excellent discrimination. Troponin-I, in contrast, did not show better discriminatory capacity in the same analysis (area under the curve 0.7733, Figure 3). Either T2-mapping at a cut-off of >50 ms for defining abnormal values or the T2-based LLc could discriminate between the two groups with a sensitivity of 96.4% and a specificity of 91.3%. The CMR findings in two illustrative cases of patients with a genetic and inflammatory myopathy are, respectively, shown in Figure 4 and Figure 5.

## 4. Discussion

In this study, we compared the CMR findings of patients with genetic and inflammatory myopathies referred for atypical cardiac symptoms to those of age- and sex-matched controls. Both myopathy groups showed largely preserved biventricular function and morphology. However, they both exhibited abnormalities in myocardial tissue characterization, with the vast majority of patients exhibiting evidence of diffuse or replacement myocardial fibrosis. Interestingly, we show that the distinguishing feature between patients with genetic and inflammatory myopathies is the presence of myocardial oedema in the latter, which could differentiate between the two groups with excellent sensitivity and specificity.

To our knowledge, this is the first study in the literature that compared the CMR findings of patients with genetic and inflammatory myopathies. Our findings show that expansion of the cardiac extracellular space, consistent with myocardial fibrosis, is a shared feature of both genetic and inflammatory myopathies. However, the fact that T2-based indices can readily discriminate between the two groups implies different etiologies leading to extracellular space expansion. Namely, increases in capillary leak leading to interstitial oedema would lead to the increases in T2-based indices observed in the inflammatory myopathy group. This corresponds with endothelial activation and subsequent capillary leak, consistent with severe acute inflammation. In contrast, although patients with genetic myopathies had, on average, significantly higher T2 mapping values compared with their matched controls, elevations comparable to those seen in patients with inflammatory myopathies were not observed. Collectively, this implies that slow, low-grade inflammation is responsible for the expansion of the cardiac extracellular space in patients with genetic myopathies. This is further supported by the beneficial effect of immunomodulatory treatment with corticosteroids in both disease groups [25,26].

The findings of this investigation are in agreement with those of previous studies. In patients with inflammatory myopathies, the main CMR pattern found is that of myocardial inflammation, with increased values of native T1 and T2 mapping and preserved LVEF [27]. Previous studies have also shown that although T2 mapping may be increased in patients with genetic myopathies, it is not a common finding [28]. In a few cases of genetic myopathies in our study, large increases in T2-based indices proportional to those of patients with inflammatory myopathies were observed. In those particular cases, the presence of abnormalities in T2-based indices should raise clinical suspicion of concomitant myocarditis.

In patients with Duchenne muscular dystrophy, LGE-spared regions showed significantly different native T1 and ECV values compared to controls [29]. The inclusion of multiple T1-based indices into a single T1-based criterion may thus increase the diagnostic yield in these cases. Previous studies have also shown the presence of increased native T1 mapping values in both asymptomatic DMD patients [29] and patients with inflammatory myopathies [27,30]. Our findings are in agreement with these studies. However, according to our findings, native T1 mapping values were higher in inflammatory compared to genetic myopathies, as expected in the context of an acute systemic inflammatory process.

### Potential Clinical and Therapeutic Implications

Our findings have important implications:(1)The presence of abnormalities in myocardial tissue characterization indices, despite the relatively preserved biventricular structure and function in patients with both genetic and inflammatory myopathies, supports the notion that disease onset may be evident even in the absence of decrements in LVEF.(2)Whether the early initiation of renin-angiotensin system inhibitors and/or β-adrenoreceptor antagonists could modulate myocardial inflammation and prevent evolution to heart failure in these patients should be investigated.(3)Gene therapy for genetic myopathies can potentially exacerbate myocardial inflammation (24). CMR could function as a screening tool before the initiation of gene therapy in these cases, but additional research is required to demonstrate where this would lead to added benefits.

## 5. Limitations

Firstly, the relatively small patient population and the single-center nature of our study should be acknowledged as limitations of this investigation. In addition, short- or long-term follow-up was not available, thus precluding the determination of the prognostic implications of CMR findings in the examined patients. Although T2 mapping could discriminate between patients with genetic and inflammatory myopathies with great accuracy, it should be acknowledged that in clinical practice, such a distinction would also be possible based on the clinical picture. Lastly, the longitudinal evaluation of myocardial inflammation and its response to targeted therapeutic interventions using CMR was not available.

## 6. Conclusions

Myocardial inflammation is a common pathway in asymptomatic patients with either genetic or inflammatory myopathies, and therefore, they should be evaluated irrespective of cause and symptoms using CMR. However, myocardial inflammation, as assessed by CMR, often has characteristics of an acute phenomenon in inflammatory myopathies while presenting with a low-grade inflammatory pattern in genetic myopathies.

## Figures and Tables

**Figure 1 jcm-12-01575-f001:**
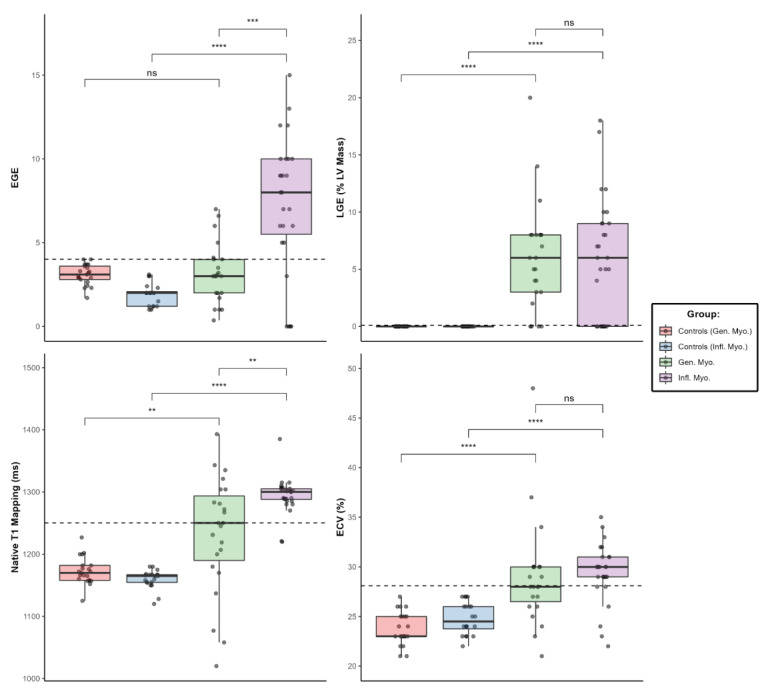
Boxplots and individual data points for T1-based CMR indices. *p*-values for individual comparisons are presented numerically in the corresponding tables (Table 1, Table 2 and Table 3). Cut-off points for defining normal values are denoted with dashed lines for each measurement. CMR: cardiovascular magnetic resonance; EGE/LGE: early/late gadolinium enhancement, ECV: extracellular volume fraction. ns: not significant; ** *p* ≤ 0.01; *** *p* ≤ 0.001, **** *p* ≤ 0.0001.

**Figure 2 jcm-12-01575-f002:**
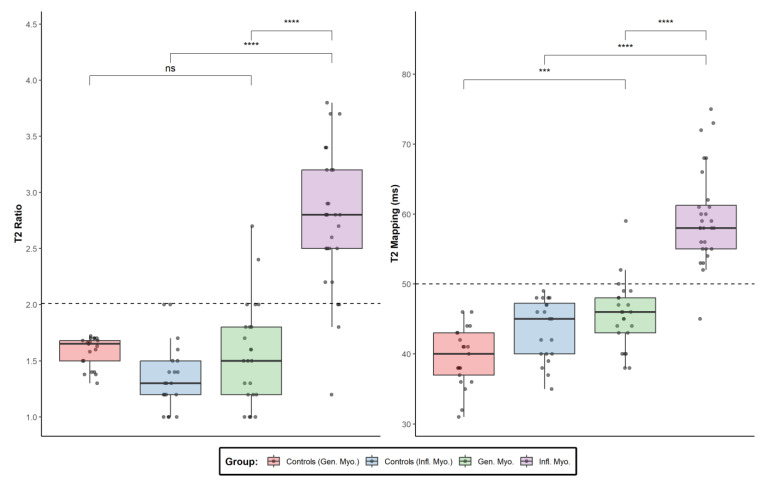
Boxplots and individual data points for T2-based CMR indices. *p*-values for individual comparisons are presented numerically in the corresponding tables (Table 1, Table 2 and Table 3). Cut-off points for defining normal values are denoted with dashed lines for each measurement. CMR: cardiovascular magnetic resonance. ns: not significant; *** *p* ≤ 0.001, **** *p* ≤ 0.0001.

**Figure 3 jcm-12-01575-f003:**
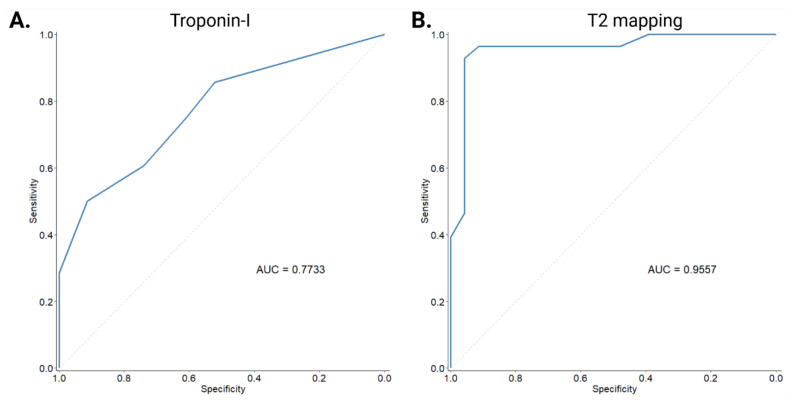
Receiver operating characteristic curve for discriminating between patients with genetic and inflammatory myopathies using troponin-I (**A**) and T2 mapping (**B**). AUC: area under the curve.

**Figure 4 jcm-12-01575-f004:**
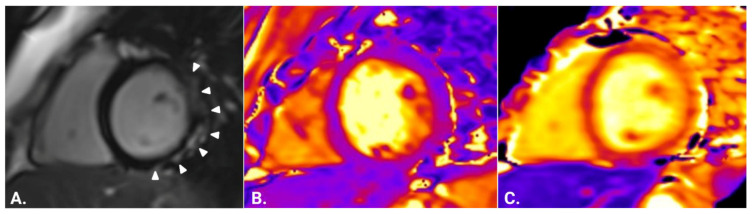
Representative cardiovascular magnetic resonance findings in a patient with Duchenne muscular dystrophy. (**A**) Short-axis inversion recovery imaging shows subepicardial fibrosis in the inferolateral wall of the left ventricle (arrowheads) (5% of left ventricular mass). (**B**) Short-axis T2 mapping of the same patient (average 45 ms). (**C**) Short-axis native T1 mapping of the same patient (average 1343 ms).

**Figure 5 jcm-12-01575-f005:**
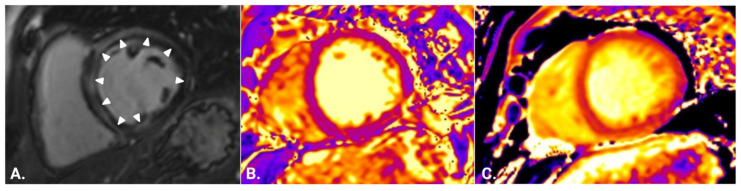
Representative cardiovascular magnetic resonance findings in a patient with polymyositis. (**A**) Short-axis inversion recovery shows extensive diffuse midwall myocardial fibrosis (arrowheads) (17% of left ventricular mass). (**B**) Short-axis T2 mapping of the same patient (average 68 ms). (**C**) Short-axis native T1 mapping of the same patient (average 1300 ms).

**Table 1 jcm-12-01575-t001:** Comparison of patients with genetic cardiomyopathy with their corresponding healthy control group. * *p* ≤ 0.05.

Variable	Matched Healthy Controls	Genetic Myopathy	*p*-Value
**Group Size**	21	23	N/A
**Demographics**			
Age (Years)	13.00 (11.00, 15.00)	12.00 (10.50, 15.00)	0.878
Female Sex (%)	10 (47.6)	5 (21.7)	0.136
**Laboratory Indices**			
Creatine Kinase (mg/L) (n < 120)	N/A	4200.00 (2750.00, 5200.00)	N/A
Troponin-I (ng/L) (n < 0.04)		0.00 (0.00, 0.03)	
**Disease Type (%)**	N/A		N/A
ANO5 Muscle Disease		1 (4.3)	
BMD		7 (30.4)	
DMD		8 (34.8)	
DMD carrier		1 (4.3)	
Friedreich Ataxia		3 (13.0)	
McLeod Syndrome		1 (4.3)	
Uknown Myopathy		2 (8.7)	
**Ventricular Structure and Function**			
LVEDVi (mL/m^2^)	67.49 (61.93, 71.32)	64.96 (54.79, 74.77)	0.488
LVESVi (mL/m^2^)	25.00 (23.03, 29.23)	25.33 (22.11, 31.25)	0.879
LVEF (%)	62.50 (59.23, 62.96)	60.27 (54.36, 63.25)	0.245
LVMi (g/m^2^)	37.76 (33.28, 43.90)	42.55 (35.08, 65.24)	0.084
RVEDVi (mL/m^2^)	60.37 (56.30, 69.03)	58.87 (44.80, 79.05)	0.445
RVESVi (mL/m^2^)	23.96 (21.97, 29.81)	27.49 (19.81, 34.18)	0.698
RVEF (%)	59.79 (57.45, 61.54)	52.46 (48.53, 62.40)	0.062
**T1-Based Indices**			
EGE	3.10 (2.80, 3.60)	3.00 (2.00, 4.00)	0.832
LGE (% of LV mass)	0.00 (0.00, 0.00)	6.00 (3.00, 8.00)	**<0.001 ***
Native T1 Mapping (ms)	1170.00 (1158.00, 1182.00)	1250.00 (1190.00, 1293.50)	**0.001 ***
Post-contrast T1 Mapping (ms)	500.00 (493.00, 523.00)	536.00 (478.50, 551.50)	0.078
ECV (%)	23.00 (23.00, 25.00)	28.00 (26.50, 30.00)	**<0.001 ***
**LGE Localization**	N/A		N/A
Anterior		2 (8.7)	
Inferior		23 (100)	
Interventricular Septum		23 (100)	
Lateral		16 (69.9)	
**T2-Based Indices**			
T2 ratio	1.65 (1.50, 1.68)	1.50 (1.20, 1.80)	0.646
T2 Mapping (ms)	40.00 (37.00, 43.00)	46.00 (43.00, 48.00)	**<0.001 ***
**Cut-off Points for Normal Values**			
EGE > 4 (%)	0 (0.0)	5 (21.7)	0.073
LGE > 0% (%)	0 (0.0)	19 (82.6)	**<0.001 ***
Native T1 Mapping > 1250 ms (%)	0 (0.0)	10 (43.5)	**0.002 ***
ECV > 28% (%)	0 (0.0)	11 (47.8)	**0.001 ***
T2 ratio > 2 (%)	0 (0.0)	2 (8.7)	0.51
T2 Mapping > 50 ms (%)	0 (0.0)	2 (8.7)	0.51
**Updated Lake Louise Criteria**			
T1 Criterion Positive (%)	0 (0.0)	22 (95.7)	**<0.001 ***
T2 Criterion Positive (%)	0 (0.0)	3 (13.0)	0.265
Both Criteria Positive (%)	0 (0.0)	3 (13.0)	0.265

N/A: not applicable; BMD: Becker muscular dystrophy; DMD: Duchenne muscular dystrophy; LVEDVi/LVESVi: indexed left ventricular end-diastolic/end-systolic volume; RVEDVi/RVESVi: indexed right ventricular end-diastolic/end-systolic volume; LVEF/RVEF: left/right ventricular ejection fraction, LVMi: indexed left ventricular mass, EGE/LGE: early/late gadolinium enhancement; ECV: extracellular volume fraction.

**Table 2 jcm-12-01575-t002:** Comparison of patients with inflammatory cardiomyopathy with their corresponding healthy control group. * *p* ≤ 0.05.

Variable	Matched Healthy Controls	Inflammatory Myopathy	*p*-Value
**Group Size**	20	28	N/A
**Demographics**			
Age (Years)	60.50 (56.25, 68.00)	60.50 (55.00, 65.00)	0.66
Female sex (%)	9 (45.0)	13 (46.4)	0.999
**Laboratory Indices**	N/A		N/A
Creatine Kinase (mg/L) (n < 120)		975.00 (780.00, 1225.00)	
Troponin-I (ng/L) (n < 0.04)		0.04 (0.02, 0.06)	
**Disease Type (%)**	N/A		N/A
Dermatomyositis		10 (35.7)	
Inclusion Body Myositis		3 (10.7)	
Myasthenia Gravis		2 (7.1)	
Polymyalgia Rheumatica		3 (10.7)	
Polymyositis		10 (35.7)	
**Ventricular Structure and Function**			
LVEDVi (mL/m^2^)	65.75 (59.28, 72.36)	74.85 (63.56, 82.64)	0.054
LVESVi (mL/m^2^)	28.13 (25.23, 31.73)	25.01 (18.53, 28.99)	0.090
LVEF (%)	58.40 (54.13, 60.20)	68.16 (61.15, 71.86)	**<0.001 ***
LVMi (g/m^2^)	57.47 (54.80, 65.17)	51.98 (48.57, 55.16)	**0.001 ***
RVEDVi (mL/m^2^)	53.22 (50.92, 61.47)	67.97 (56.06, 77.26)	**0.007 ***
RVESVi (mL/m^2^)	23.26 (20.45, 25.84)	24.72 (23.41, 32.28)	0.149
RVEF (%)	57.89 (53.96, 59.39)	58.01 (54.51, 65.30)	0.477
**T1-Based Indices**			
EGE	2.00 (1.20, 2.08)	8.00 (5.50, 10.00)	**<0.001 ***
LGE (% of LV mass)	0.00 (0.00, 0.00)	6.00 (0.00, 9.00)	**<0.001 ***
Native T1 Mapping (ms)	1165.00 (1154.75, 1167.00)	1300.00 (1288.00, 1305.00)	**<0.001 ***
Post-contrast T1 Mapping (ms)	490.00 (480.00, 516.25)	530.00 (499.00, 548.50)	**0.005 ***
ECV (%)	24.50 (23.75, 26.00)	30.00 (29.00, 31.00)	**<0.001 ***
**LGE Localization**	N/A		N/A
Anterior		2 (7.1)	
Inferior		28 (100)	
Interventricular Septum		3 (10.7)	
Lateral		20 (71.4)	
**T2-Based Indices**			
T2 ratio	1.30 (1.20, 1.50)	2.80 (2.50, 3.20)	**<0.001 ***
T2 Mapping (ms)	45.00 (40.00, 47.25)	58.00 (55.00, 61.25)	**<0.001 ***
**Cut-off Points for Normal Values**			
EGE > 4 (%)	0 (0.0)	22 (81.5)	**<0.001 ***
LGE > 0% (%)	0 (0.0)	19 (67.9)	**<0.001 ***
Native T1 Mapping > 1250 ms (%)	0 (0.0)	26 (92.9)	**<0.001 ***
ECV > 28% (%)	0 (0.0)	22 (78.6)	**<0.001 ***
T2 ratio > 2 (%)	0 (0.0)	24 (85.7)	**<0.001 ***
T2 Mapping > 50 ms (%)	0 (0.0)	27 (96.4)	**<0.001 ***
**Updated Lake Louise Criteria**			
T1 Criterion Positive (%)	0 (0.0)	28 (100.0)	**<0.001 ***
T2 Criterion Positive (%)	0 (0.0)	27 (96.4)	**<0.001 ***
Both Criteria Positive (%)	0 (0.0)	27 (96.4)	**<0.001 ***

LVEDVi/LVESVi: indexed left ventricular end-diastolic/end-systolic volume; RVEDVi/RVESVi: indexed right ventricular end-diastolic/end-systolic volume; LVEF/RVEF: left/right ventricular ejection fraction, LVMi: indexed left ventricular mass, EGE/LGE: early/late gadolinium enhancement; ECV: extracellular volume fraction.

**Table 3 jcm-12-01575-t003:** Comparison of patients with genetic and inflammatory myopathies. * *p* ≤ 0.05.

Variable	Genetic Myopathy	Inflammatory Myopathy	*p*-Value
**Group Size**	23	28	N/A
**Demographics**			
Age (Years)	12.00 (10.50, 15.00)	60.50 (55.00, 65.00)	**<0.001 ***
Female Sex (%)	5 (21.7)	13 (46.4)	0.123
**Laboratory Indices**			
Creatine Kinase (mg/L) (n < 120)	4200.00 (2750.00, 5200.00)	975.00 (780.00, 1225.00)	**<0.001 ***
Troponin-I (ng/L) (n < 0.04)	0.00 (0.00, 0.03)	0.04 (0.02, 0.06)	**0.001 ***
**Ventricular Structure and Function**			
LVEDVi (mL/m^2^)	64.96 (54.79, 74.77)	74.85 (63.56, 82.64)	0.085
LVESVi (mL/m^2^)	25.33 (22.11, 31.25)	25.01 (18.53, 28.99)	0.394
LVEF (%)	60.27 (54.36, 63.25)	68.16 (61.15, 71.86)	**<0.001 ***
LVMi (g/m^2^)	42.55 (35.08, 65.24)	51.98 (48.57, 55.16)	0.135
RVEDVi (mL/m^2^)	58.87 (44.80, 79.05)	67.97 (56.06, 77.26)	0.198
RVESVi (mL/m^2^)	27.49 (19.81, 34.18)	24.72 (23.41, 32.28)	0.835
RVEF (%)	52.46 (48.53, 62.40)	58.01 (54.51, 65.30)	0.066
**T1-Based Indices**			
EGE	3.00 (2.00, 4.00)	8.00 (5.50, 10.00)	**<0.001 ***
LGE (% of LV mass)	6.00 (3.00, 8.00)	6.00 (0.00, 9.00)	0.924
Native T1 Mapping (ms)	1250.00 (1190.00, 1293.50)	1300.00 (1288.00, 1305.00)	**0.004 ***
Post-contrast T1 Mapping (ms)	536.00 (478.50, 551.50)	530.00 (499.00, 548.50)	0.805
ECV (%)	28.00 (26.50, 30.00)	30.00 (29.00, 31.00)	0.128
**LGE Localization**			
Anterior	2 (8.7)	2 (7.1)	0.999
Inferior	23 (100)	28 (100)	0.999
Interventricular Septum	0 (0)	3 (10.7)	0.308
Lateral	16 (69.6)	20 (71.4)	0.999
**T2-Based Indices**			
T2 ratio	1.50 (1.20, 1.80)	2.80 (2.50, 3.20)	**<0.001 ***
T2 Mapping (ms)	46.00 (43.00, 48.00)	58.00 (55.00, 61.25)	**<0.001 ***
**Cut-off Points for Normal Values**			
EGE > 4 (%)	5 (21.7)	22 (81.5)	**<0.001 ***
LGE > 0% (%)	19 (82.6)	19 (67.9)	0.379
Native T1 Mapping > 1250 ms (%)	10 (43.5)	26 (92.9)	**<0.001 ***
ECV > 28% (%)	11 (47.8)	22 (78.6)	**0.046 ***
T2 ratio > 2 (%)	2 (8.7)	24 (85.7)	**<0.001 ***
T2 Mapping > 50 ms (%)	2 (8.7)	27 (96.4)	**<0.001 ***
**Updated Lake Louise Criteria**			
T1 Criterion Positive (%)	22 (95.7)	28 (100.0)	0.921
T2 Criterion Positive (%)	3 (13.0)	27 (96.4)	**<0.001 ***
Both Criteria Positive (%)	3 (13.0)	27 (96.4)	**<0.001 ***

LVEDVi/LVESVi: indexed left ventricular end-diastolic/end-systolic volume; RVEDVi/RVESVi: indexed right ventricular end-diastolic/end-systolic volume; LVEF/RVEF: left/right ventricular ejection fraction, LVMi: indexed left ventricular mass, EGE/LGE: early/late gadolinium enhancement; ECV: extracellular volume fraction.

## Data Availability

The data used in this study can be made available upon reasonable request to the corresponding author.

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
