# Peer review of "Cardiovascular Magnetic Resonance Demonstrates Myocardial Inflammation of Differing Etiologies and Acuities in Patients with Genetic and Inflammatory Myopathies"

_jcm, 2023, doi:10.3390/jcm12041575_

Round 1
Reviewer 1 Report
In their manuscript, “Cardiovascular Magnetic Resonance Demonstrates Myocardial Inflammation of Differing Etiologies and Acuities in Patients with Genetic and Inflammatory Myopathies,” the authors investigate CMR evidence of cardiac involvement in patients with genetic and inflammatory extra-cardiac myopathies. The study arm included 23 patients with genetic extra-cardiac myopathy and 28 patients with inflammatory extra-cardiac myopathy, all with atypical cardiac symptoms and normal echo. Genetic myopathies were diverse, with Becker’s and Duchene’s muscular dystrophy being the most common, accounting for a combined ~65%. Inflammatory myopathies were also diverse, with dermatomyositis and polymyositis being the most common, accounting for a combined ~71%. CMR exam included bSSFP cine, STIR-T2, LGE, and MOLLI, with ECV reconstruction, which is an appropriately broad protocol. bSSFP cine images were used to quantify biventricular ESV, EDV, and EF as well as LV mass. Compared to control subjects, patients with genetic myopathies had similar biventricular morphology, function, and STIR-T2, but higher LGE, T1, ECV, and T2. In particular, 95.7% met LLc T1-based criteria. Compared to control subjects, patients with inflammatory myopathies had preserved LVF, reduced LV mass, and all tissue characterization indices were higher. In particular, 100% of patients met LLc T1-based criteria, and 96.4% met T2-based criteria.
The manuscript is well written, well organized, and thoroughly copy-edited. The topic is interesting and certainly of interest to the readers of this journal. The scientific methodology is sound. The primary takeaways are that (a) CMR showed a high prevalence of cardiac involvement in patients with genetic and inflammatory extra-cardiac myopathies and that (b) CMR can be used to quite reliably distinguish the two. This result is clinically useful, consistent with prior literature, and sufficiently novel. Limitations are appropriately discussed. I have no major concerns with the manuscript. Minor concerns are as follows:
Minor Concerns:
1. Characterization of the myopathy subtypes is appropriately detailed in the tables, but could be briefly discussed in the results as well.
2. What matching procedure was used?
Author Response
In their manuscript, “Cardiovascular Magnetic Resonance Demonstrates Myocardial Inflammation of Differing Etiologies and Acuities in Patients with Genetic and Inflammatory Myopathies,” the authors investigate CMR evidence of cardiac involvement in patients with genetic and inflammatory extra-cardiac myopathies. The study arm included 23 patients with genetic extra-cardiac myopathy and 28 patients with inflammatory extra-cardiac myopathy, all with atypical cardiac symptoms and normal echo. Genetic myopathies were diverse, with Becker’s and Duchene’s muscular dystrophy being the most common, accounting for a combined ~65%. Inflammatory myopathies were also diverse, with dermatomyositis and polymyositis being the most common, accounting for a combined ~71%. CMR exam included bSSFP cine, STIR-T2, LGE, and MOLLI, with ECV reconstruction, which is an appropriately broad protocol. bSSFP cine images were used to quantify biventricular ESV, EDV, and EF as well as LV mass. Compared to control subjects, patients with genetic myopathies had similar biventricular morphology, function, and STIR-T2, but higher LGE, T1, ECV, and T2. In particular, 95.7% met LLc T1-based criteria. Compared to control subjects, patients with inflammatory myopathies had preserved LVF, reduced LV mass, and all tissue characterization indices were higher. In particular, 100% of patients met LLc T1-based criteria, and 96.4% met T2-based criteria.
The manuscript is well written, well organized, and thoroughly copy-edited. The topic is interesting and certainly of interest to the readers of this journal. The scientific methodology is sound. The primary takeaways are that (a) CMR showed a high prevalence of cardiac involvement in patients with genetic and inflammatory extra-cardiac myopathies and that (b) CMR can be used to quite reliably distinguish the two. This result is clinically useful, consistent with prior literature, and sufficiently novel. Limitations are appropriately discussed. I have no major concerns with the manuscript.
We thank the reviewer for the kind appraisal of our work.
Minor Concerns:
- Characterization of the myopathy subtypes is appropriately detailed in the tables, but could be briefly discussed in the results as well.
We thank the reviewer for this remark. We now state in the results the major disease subtypes in each group as follows:
“The majority of patients with genetic myopathies were diagnosed either with Duchenne muscular dystrophy [8 (34.8%)] or Becker muscular dystrophy [7 (30.4%)].”
“The majority of patients with inflammatory myopathies were diagnosed either with polymyositis [10 (35.7%)] or dermatomyositis [10 (35.7%)].”
- What matching procedure was used?
Two separate control groups were matched to the cohorts of patients with genetic and inflammatory myopathies. This was done because patients in the genetic myopathy group were either children or adolescents, which was not the case in the inflammatory myopathy group. Thus, matching of healthy controls was performed separately for each group based on sex and age-matching (using descriptive statistics).
Reviewer 2 Report
The authors of this work study patients with myopathy on cardiac MRI, looking for signs of inflammation or fibrosis, and compared their results with controlled subjects.
The results show a frequent cardiac edema in inflammatory heart diseases, rarer in genetic cardiomyopathies which present rather fibrous changes. The T2 map seems effective in separating the 2 types of myopathies.
The article brings a lot of data in MRI. Those on inflammatory myopathies are globally already known. The presence of myocardial edema in genetic myopathies has been less reported and is of interest.
The authors use the T2 map to separate the 2 types of myopathies via a ROC curve. This parameter (T2) obviously works but it seems a very theoretical comparison because, in clinical practice, the question of the differential diagnosis between these 2 types of myopathies, quite different clinically, does not really arise, unless I am mistaken. The usefulness of T2 to separate these 2 diseases therefore seems quite theoretical because, as mentioned below, the clinical context is different. This point should be addressed in the discussion or limitations.
In the description of the patients, one should also give biological data such as the value of troponin and muscle enzymes. If the authors maintain the ROC curve of T2 to separate patients, it should be compared to that of troponin.
The conclusion of the abstract seems wrong to me: the inflammatory component is not common in genetic myopathies, it even seems rather rare given the reported data. On the other hand, it is frequent in inflammatory myopathies.
Points 2 and 3 in the clinical implications should be deleted: the data contained here do not support these therapeutic hypotheses
Minor remarks
Use the term T2 ratio rather than T2 STIR for the heart T2 / muscle T2 ratio.
The location of the LGE anomalies is not specified in the results unless I am mistaken.
Give the value of the patients' T2 and T1 map for figures A and B
Author Response
The authors of this work study patients with myopathy on cardiac MRI, looking for signs of inflammation or fibrosis, and compared their results with controlled subjects. The results show a frequent cardiac edema in inflammatory heart diseases, rarer in genetic cardiomyopathies which present rather fibrous changes. The T2 map seems effective in separating the 2 types of myopathies. The article brings a lot of data in MRI. Those on inflammatory myopathies are globally already known. The presence of myocardial edema in genetic myopathies has been less reported and is of interest.
We thank the reviewer for the kind appraisal of our work.
The authors use the T2 map to separate the 2 types of myopathies via a ROC curve. This parameter (T2) obviously works but it seems a very theoretical comparison because, in clinical practice, the question of the differential diagnosis between these 2 types of myopathies, quite different clinically, does not really arise, unless I am mistaken. The usefulness of T2 to separate these 2 diseases therefore seems quite theoretical because, as mentioned below, the clinical context is different. This point should be addressed in the discussion or limitations.
We have now added the following sentence to the limitations section, based on the reviewer’s suggestion:
“Although T2 mapping could discriminate between patients with genetic and inflammatory myopathies with great accuracy, it should be acknowledged that in clinical practice such a distinction would also be possible based on the clinical picture.”
In the description of the patients, one should also give biological data such as the value of troponin and muscle enzymes. If the authors maintain the ROC curve of T2 to separate patients, it should be compared to that of troponin.
The values of creatine kinase and troponin-I has now been added for patients with inflammatory and genetic myopathies in Tables 1, 2 and 3. We have also added an ROC curve for Troponin-I in Figure 3 and state the following in the text:
“Troponin-I, in contrast, did not show better discriminatory capacity in the same analysis (area under the curve 0.7733, Figure 3).”
The conclusion of the abstract seems wrong to me: the inflammatory component is not common in genetic myopathies, it even seems rather rare given the reported data. On the other hand, it is frequent in inflammatory myopathies.
We thank the reviewer for pointing this out. We have now rephrased the conclusion of the abstract as follows:
“Conclusions. The vast majority of symptomatic patients with inflammatory myopathies and normal echocardiography show evidence of acute myocardial inflammation. In contrast, acute inflammation is rare in patients with genetic myopathies, who show evidence chronic low-grade inflammation.”
Points 2 and 3 in the clinical implications should be deleted: the data contained here do not support these therapeutic hypotheses.
We thank the reviewer for this remark. We have now rephrased the implications section and provided a new title: “Potential Clinical and Therapeutic Implications”.
The text (particularly points 2 and 3) has now been edited and is now more conservatively phrased. We also explicitly mention that points 2 and 3 need to be further investigated in the future. The text now reads as follows:
Our findings have important implications:
1) The presence of abnormalities in myocardial tissue characterization indices, despite the relatively preserved biventricular structure and function in patients with both genetic and inflammatory myopathies, supports the notion that disease onset may be evident even in the absence of decrements in LVEF.
2) Whether the early initiation of renin-angiotensin system-inhibitors and/or β-adrenoreceptor antagonists could modulate myocardial inflammation and prevent evolution to heart failure in these patients should be investigated.
3) Gene therapy for genetic myopathies can potentially exacerbate myocardial inflammation (24). CMR could function as a screening tool before initiation of gene therapy in these cases, but additional research is required to demonstrate where this would lead to added benefits.
Minor remarks
Use the term T2 ratio rather than T2 STIR for the heart T2 / muscle T2 ratio.
Thank you, this has now been replaced throughout the text with “T2 ratio”.
The location of the LGE anomalies is not specified in the results unless I am mistaken.
The localization of LGE anomalies has now been added in Tables 1, 2 and 3.
Give the value of the patients' T2 and T1 map for figures A and B
Please find below the value for LGE and mapping indices for figures 4 and 5. These have also been added to the corresponding figure legends.
Figure 4: LGE = 5%, T2 mapping = 45 ms, native T1 mapping = 1343 ms.
Figure 5: LGE = 17%, T2 mapping = 68 ms, native T1 mapping = 1300 ms.